# Antifungal Activity and In Silico Studies on 2-Acylated Benzo- and Naphthohydroquinones

**DOI:** 10.3390/molecules27093035

**Published:** 2022-05-09

**Authors:** David Ríos, Jaime A. Valderrama, Gonzalo Quiroga, Jonathan Michea, Felipe Salas, Eduardo Álvarez Duarte, Edmundo A. Venegas-Casanova, Rafael Jara-Aguilar, Carlos Navarro-Retamal, Pedro Buc Calderon, Julio Benites

**Affiliations:** 1Química y Farmacia, Facultad de Ciencias de la Salud, Universidad Arturo Prat, Casilla 121, Iquique 1100000, Chile; darios@unap.cl (D.R.); jaimeadolfov@gmail.com (J.A.V.); gonzalo.quiroga1024@gmail.com (G.Q.); jmichea25@gmail.com (J.M.); felisala@unap.cl (F.S.); pedro.buccalderon@uclouvain.be (P.B.C.); 2Unidad de Micología, Instituto de Ciencias Biomédicas, Facultad de Medicina, Universidad de Chile, Av. Independencia 1027, Santiago 8380453, Chile; ealvarezd@med.uchile.cl; 3Facultad de Farmacia y Bioquímica, Universidad Nacional de Trujillo, Trujillo 13011, Peru; evenegas@unitru.edu.pe (E.A.V.-C.); djara@unitru.edu.pe (R.J.-A.); 4Instituto de Ciencias Biológicas, Universidad de Talca, 2 Norte 685, Talca 3460000, Chile; cnavarro@utalca.cl; 5Research Group in Metabolism and Nutrition, Louvain Drug Research Institute, Université Catholique de Louvain, 73 Avenue E. Mounier, 1200 Brussels, Belgium

**Keywords:** antifungal activity, *Candida*, filamentous fungi, acylhydroquinones

## Abstract

The high rates of morbidity and mortality due to fungal infections are associated with a limited antifungal arsenal and the high toxicity of drugs. Therefore, the identification of novel drug targets is challenging due to the several resemblances between fungal and human cells. Here, we report the in vitro antifungal evaluation of two acylphenols series, namely 2-acyl-1,4-benzo- and 2-acyl-1,4-naphthohydroquinones. The antifungal properties were assessed on diverse *Candida* and filamentous fungi strains through the halo of inhibition (HOI) and minimal inhibitory concentration (MIC). The antifungal activities of 2-acyl-1,4-benzohydroquinone derivatives were higher than those of the 2-acyl-1,4-naphthohydroquinone analogues. The evaluation indicates that 2-octanoylbenzohydroquinone **4** is the most active member of the 2-acylbenzohydroquinone series, with MIC values ranging from 2 to 16 μg/mL. In some fungal strains (i.e., *Candida krusei* and *Rhizopus oryzae*), such MIC values of compound **4** (2 and 4 μg/mL) were comparable to that obtained by amphotericin B (1 μg/mL). The compound **4** was evaluated for its antioxidant activity by means of FRAP, ABTS and DPPH assays, showing moderate activity as compared to standard antioxidants. Molecular docking studies of compound **4** and ADMET predictions make this compound a potential candidate for topical pharmacological use. The results obtained using the most active acylbenzohydroquinones are promising because some evaluated *Candida* strains are known to have decreased sensitivity to standard antifungal treatments.

## 1. Introduction

A dramatic increase in the frequency of infections by opportunistic fungi, mainly mediated by *Candida* (Saccharomycetales) and *Aspergillus* (Eurotiales) species, including invasive fungal infections (IFIs), is reported in immunocompromised patients [1,2,3,4,5]. Although infections by filamentous fungi are less frequent than infection by *Candida* spp., they are associated with higher mortality rates [6,7]. Candidiasis is a primary or secondary infection caused by yeasts belonging to the genus *Candida*. Among them, *Candida albicans* is the pathogen species usually found in clinical samples, being the etiological agent in most cases of vulvovaginitis [8,9]. Recent reports of isolation of other different *Candida* species such as *Candida glabrata*, *Candida parapsilosis*, and *Candida tropicalis* significantly increased the number of associated infections [10,11]. Over the past few decades, the increased use of antifungal drugs (fluconazole and other triazoles) against candidiasis has led to the development of drug resistance in several fungal species [12]. Therefore, the discovery of new antifungal agents is of utmost importance to provide alternative pharmacotherapy to fight against infections as well as drug resistance.

Natural phenolic compounds are organic substances that have one or more hydroxyl groups on the aromatic skeleton. They play many significant roles in human health, as evident from their antifungal [13], antioxidant [14,15,16] and anticancer activities [17]. Among the broad variety of natural antifungal phenolic compounds, the natural acylphenols **I**–**III** (Figure 1), isolated from leaves of *Piper crassinervium*, are of interest [18]. The antifungal evaluation of these compounds against *Candida cladosporioides* and *Candida sphaerospermum* reveals that acylphenol **I** exhibits the highest activities, comparable to nystatin and miconazole, two antifungal standard drugs used as positive controls.

In addition, the synthetic phenols acylbenzohydroquinone **IV** and acylnaphthohydroquinone **V** have shown in vitro antiproliferative activities against some cancer cell lines (i.e., prostate, bladder, breast). Further biological studies with synthetic aroylnaphthohydroquinones revealed a high affinity of analogue **VI** to inhibit microtubule assembly [19,20].

Since the natural prenylated acylhydroquinones **I**–**III** (Figure 1) display antifungal activity, structurally related synthetic acylhydroquinones were studied. To this end, the potential antifungal activity of a representative number of members of the acylbenzo- and acylnaphthohydroquinones series toward *Candida* spp. (*Candida albicans*, *Candida glabrata*, *Candida krusei*) and some filamentous fungal species (*Aspergillus fumigatus*, *Aspergillus terreus*, *Acremonium kiliense*, *Rhizopus oryzae*) were assessed. We performed additional assays in order to understand the mechanisms explaining the potential antifungal activity as well as investigating some molecular targets. They included antioxidant capacity, in silico molecular docking analysis against *Candida albicans* NADPH dehydrogenase, and the molecular dynamic and pharmacokinetic properties of an active member of the acylphenol series.

## 2. Results

### 2.1. Antifungal Activity, Halo of Inhibition (HOI)

Table 1 shows the resulting diameters of the halo of inhibition (HOI) for *Candida* species and filamentous fungi obtained after exposition for 24 h (yeasts) and 48 h (molds) to acylhydroquinones at 50 ppm. Evaluation of yeasts included *Candida albicans*, *Candida glabrata* and *Candida krusei*; while filamentous fungi included *Aspergillus fumigatus* 280 (an azole-resistant strain), *Aspergillus fumigatus* 2H, *Aspergillus terreus*, *Acremonium kiliense* and *Rhizopus oryzae*. Note that *Candida* species were largely more sensitive than filamentous fungi. Indeed, 85% of the molecules (12/14) had antifungal activity against at least two to three *Candida* species, while only three molecules were active against the four species of filamentous fungi, including both *Aspergillus fumigatus* strains.

With regard to *Candida glabrata*, both acylhydroquinones displayed the largest HOI, while *Aspergillus fumigatus* 2H strain was the most sensitive among the filamentous fungi. In general, the antifungal activities of 2-acylbenzohydroquinone derivatives (2-ABHQ) were more potent than 2-acylnaphthohydroquinone derivatives (2-ANHQ). Interestingly, compounds **3** and **4** were active against both fungal types, while their naphthohydroquinone derivatives **10** and **11** were devoid of antifungal activity in almost all tested species (minimal activity was seen with **10** but only in *Candida* species). 

### 2.2. Determination of Minimal Inhibitory Concentration (MIC) Values

Compounds displaying the best activity (high HOI values) were selected to calculate MIC values. They included the benzoyl derivatives **3**, **4** and **5**, and the naphthyl derivatives **12**, **14** and **15**. Such activity was further compared to that observed with amphotericin B, a standard antifungal drug. This drug binds ergosterol, a component of fungal cell membranes, forming pores that cause leakage of cellular components, leading to fungal cell death [21]. Table 2 summarizes the MIC values of amphotericin B and the selected acylhydroquinones. 

The results show that **3** and **4** were the most active compounds against both yeasts and molds displaying MIC values in the range of 2 and 16 µg/mL in *Candida* species and 4 and 64 µg/mL in filamentous fungi. Regarding some fungal strains (i.e., *Candida krusei* 118), the MIC values of compound **4** (2 µg/mL) are close to that obtained for amphotericin B (1 µg/mL).

### 2.3. Antioxidant Capacity of Octanoylbenzohydroquinone ***4***

Due to its high antifungal activity and the antioxidant property of natural phenolic compounds [22,23,24], compound **4** was selected to further studies. To this end, its potential in vitro antioxidant activity was determined according to the in vitro FRAP, ABTS^•+^ and DPPH methods and the results are summarized in Table 3. The FRAP assay is based on the reduction of a colorless Fe^3+^-TPTZ complex into intense blue Fe^2+^-TPTZ once it interacts with a potential antioxidant. The ABTS assay is based on cation radical scavenging activity. Finally, the DPPH assay is based on the ability of a compound to act as donor for hydrogen atoms or electrons in the transformation of DPPH• into its reduced form DPPH-H. The results are displayed as the mean ± SD of triplicate tests. The well-known antioxidants quercetin and Trolox^®^ were used as positive controls. 

Irrespective of the utilized test, the results reveal the moderate antioxidant activity of compound **4**. Indeed, in the FRAP reduction method, its TEAC value was 0.30 ± 0.16 mM, a value at least six times less potent than the values of the reference standard, quercetin (TEAC ≤ 0.05 ± 0.001 mM). A similar profile was observed with the other methods (ABTS and DPPH). When compared to Trolox^®^, compound **4** was 10- to 13-fold less potent.

### 2.4. Molecular Docking of the Octanoylbenzohydroquinone ***4***

To get a better understanding about the biological pathway by which compound **4** acts as an antifungal, we explored how its molecular structure may be involved in *Candida albicans* growth inhibition. We examined, by molecular docking, the binding modes of this ligand in the enzyme ‘Old Yellow Enzyme’ (OYE1) of *Candida albicans* (code 1BWK in the Protein Data Bank). This target was selected because OYE1 is an NADPH dehydrogenase responsible for protection against oxidative stress induced by adverse environmental conditions [25,26]. Such an approach was initially tested by re-docking the molecule flavin mononucleotide, present in the active site of OYE1. The docked structure showed a RMSD of 0.3826 Å with respect to the co-crystalized molecule, supporting our approach. 

Prior to docking experiments, the binding site of *Candida albicans* NADPH dehydrogenase using Sitemap [27,28] was studied under the Schrödinger suite of programs [29]. Figure 2 shows that the pocket at the region of the binding site of OYE1 can be divided in three regions—two charged regions (named P1 and P2 regions) joined by a hydrophobic section (named H region). 

Furthermore, the residues His191, Asn194 and Gln114 of region P1 stabilize the hydroquinone fragment of the compound **4** by means of coulombic and polar interactions (Figure 3). His191 and Asn194 have been observed to be crucial for ligand binding [30], but no evidence has been found for Gln114. Moreover, multiple hydrophobic interactions formed by the hydrophobic tail of the compound with respect to the residues Leu32 and Ile351 of the region H also help to keep the structure stable in the binding pocket. The molecule was not able to interact with the residues at the region P2. The score docking of compound **4** was −4.481 Kcal/mol.

To describe the nature of the interactions between NADPH dehydrogenase and the ligands over time, a molecular dynamic (MD) simulation of the complex NADPH-octanoylbenzohydroquinone **4** was conducted. Here, three independent replicas were performed, for 100 ns each. During the simulations, intermolecular interaction between the 5-OH group of ligand **4** and the residues at region P1 was observed. This interaction is probably responsible for keeping the ligand in the binding site of the NADPH (Figure 4 Top). Interestingly, an intramolecular H-bond was observed between the 2-OH and C=O groups involving the alkanoyl substituent of the molecule (Figure 4 Bottom). Such intramolecular interactions between the 2-OH and C=O groups in compound **4** are clearly evidenced by H- and C-NMR spectroscopy. To further illustrate the interactions between the protein and the molecule, free energy calculations by means of the MM-GBSA approach were performed.

Here, it was observed that the structural key components responsible for inhibitor stabilization at the binding site of NADPH dehydrogenase are related to coulombic and polar interactions of the hydroquinone fragment and lipophilicity and ***vdW***-interactions of the C-2 alkanoyl substituent (Table 4), supporting our docking and in vitro experiments. 

### 2.5. ADMET Profiles

Table 5 shows pharmacokinetic (absorption, distribution, metabolism, excretion), and toxicity values of compound **4** and amphotericin B. They were calculated by using the pkCSM Online Tool.

The penetrability of compound **4** is promising because it has a molecular weight of 236 g/mol, a value that is lower than 500 g/mol, which is the upper limit concerning this parameter [31]. Since compound **4** had a high intestinal absorption (>90%) and Caco-2 permeability values above 1.30; this would predict its absorption in the small intestine [32]. Furthermore, the skin permeability of compound **4** was −3.125 cm/h (<−2.5), showing transdermal efficacy; that is, it will penetrate the skin properly. Indeed, it is known that molecules will have impediments in skin penetration if the log Kp value is higher than −2.5 cm/h [33]. 

Compound **4** has a suitable volume distribution (VDss) value, which is larger than −0.15 [34]; however, the blood–brain barrier (BBB) permeability will not be favored (log BB > 0.2). Conversely to BBB permeability, compound **4** has a logP value of −2.386, which is higher than −2, enabling its central nervous system (CNS) permeability [35]. 

Regarding putative inhibition or interactions with enzymes involved in xenobiotic metabolism, Table 5 shows that neither compound **4** nor amphotericin B were inhibitors of CYP2D6 and CYP3A4, suggesting no interferences with CYP450 biotransformation in general. 

In terms of excretion parameters, it is shown that compound **4** achieved positive value of total clearance (1.22 mL/min/kg), meaning a quick excretion compared with amphotericin B [36]. The adverse interactions of compound **4** with OCT2 inhibitors revealed no potential contraindication. 

Regarding the acute oral toxicity in rats, the results show that compound **4** had a LD_50_ of 1.968 mol/kg, suggesting that it is more toxic than amphotericin B (2.518 mol/kg). Additionally, compound **4** was not considered a hepatotoxic substance.

## 3. Discussion

We have observed that chemical structures of acylhydroquinones, and in particular the acyl and aromatic ring fragments, influence antifungal activity. Given that **4** (and **3** to a lesser extent) is the most promising candidate, a Structure–Activity Relationship (SAR) analysis leads to the following conclusions. Briefly, compounds **3**, **4** and **5** are active against both *Candida* and filamentous fungi spp. when considering the agar diffusion test. The rest of acylhydroquinones (compounds **6**, **7** and **8**) and all acylnaphthohydroquinone derivatives are mainly active on *Candida* strains. In general, acylbenzohydroquinone derivatives have more antifungal activity than acylnaphthohydroquinone derivatives, excepting compound **15**, which is more active than compound **7** in *Candida* species. When R is substituted by aliphatic chains, a better antifungal activity is obtained compared to the aromatic or heteroaromatic rings. Notably, the length of the aliphatic chain plays a role with regard to *Candida* species, because compound **9** (-CH_3_) is more active than **10** (-C_5_H_11_) and **11** (-C_7_H_15_). In this context, it has been reported that 1,4-benzoquinone and hydroquinone derivatives display MIC values in *Candida albicans* higher than 12.5 µg/mL [37,38]. By performing experiments with other hydroquinone and 1,4-quinone derivatives, some authors have reported a MIC higher than 50 µg/mL in *Candida albicans* strains [39,40], a rather high concentration compared to results obtained in this study.

Regarding compound **4**, it has low MIC values in both *Candida* and filamentous fungi and notably the lowest MIC value (2 µg/mL) in *Candida krusei* strains, which was even comparable to the MIC value calculated for the reference drug (amphotericin B; 1 µg/mL). This antifungal activity is important because *Candida krusei* has an intrinsic resistance to fluconazole as well as a reduced susceptibility to flucytosine and amphotericin B [41], two of the main drugs currently employed in antifungal therapy. For such reasons, *Candida krusei* has been recognized as a potentially multidrug-resistant opportunistic fungal pathogen [42]. 

To explore intracellular targets, we focused our study on NADPH dehydrogenase of *Candida albicans.* Indeed, in addition to its biological role, the enzyme binds phenolic ligands, forming long wavelength (500–800 nm) charge-transfer complexes. NADPH reduces the enzyme, and oxygen-containing compounds can act as electron acceptors to complete catalytic turnover [30]. Therefore, its inhibition could induce a significant in vitro antioxidant effect. Furthermore, by using a protocol combining molecular docking, MD simulations and free energy calculations, the interaction of the most active compound on the active site of OYE1 NADPH dehydrogenase was investigated. We observed that coulombic interactions (considering polar, as well as H-bond interactions) and van der Waal’s force (*vdW*) interactions are responsible for a proper binding of inhibitor.

Furthermore, we studied selected pharmacokinetics parameters with regard to compound **4**. Based on its structure, such ADMET results show a positive response for penetrability and Caco-2 permeability, a negative response for blood–brain barrier permeability and low toxicity. Therefore, new modified molecules made from compound **4** are expected to be safe for topical pharmacological use. 

Taken together, these findings suggest that molecular interactions between NADPH dehydrogenase and compound **4** are likely. Therefore, such interactions may be further used as templates for new drug designs targeting NADPH dehydrogenase as an original inhibitory mechanism to reach antifungal activity.

## 4. Materials and Methods

### 4.1. Synthesis of Acylhydroquinone Derivatives ***3***–***16***

The products required for the antifungal evaluation were synthesized according to our previously reported procedure, based on the solar photoacylation Friedel–Crafts reaction of quinones **1** and **2** with aldehydes [43] (Figure 1). The acylbenzohydroquinones **3**–**8** were synthesized from 1,4-benzoquinone **1** and the following aldehydes: *n*-hexanal, *n*-octanal, benzaldehyde, cinnamaldehyde, 2-thienylcarbaldehyde and 2-furylcarbaldehyde. The acylnaphthohydroquinones **9**–**16** were prepared from 1,4-naphthoquinone **2** and the following aldehydes: ethanal, *n*-hexanal, *n*-octanal, benzaldehyde, cinnamaldehyde, 3-methoxy-4-hydroxybenzaldehyde, 2-thienylcarbaldehyde and 2-furylcarbaldehyde.

### 4.2. Biological Evaluation

#### 4.2.1. Strains

The following fungal strains were employed in the screening assays: *Candida albicans* A8, *Candida glabrata* 2001 and *Candida krusei* 118; *Aspergillus fumigatus* 280 (azole resistant strain), *Aspergillus fumigatus* 2H, *Aspergillus terreus*, *Acremonium kiliense*, and *Rhizopus oryzae*, arising from the Chilean Fungal Collection (ChFC). All strains were incubated in Sabouraud dextrose agar (SDA) without antimicrobial agents at 37 °C for 3–7 days to ensure their purity and viability. Note that *Candida glabrata* and *Candida krusei* are recognized as species with low MIC values for several antifungal agents, such as flucytosine, amphotericin B, fluconazole, itraconazole, caspofungin [11,41,42,44].

#### 4.2.2. Antifungal Susceptibility Testing

Agar diffusion tests were performed using acylhydroquinones at 50 ppm. They were directly used without dilutions against the fungal (molds/yeasts) strains. Only those compounds that showed activity in this first assay advanced to the broth microdilution test, where the minimum inhibitory concentration was obtained. Results were expressed in millimeters. 

For microdilution broth methodologies, serial dilutions of each compound were performed to calculate the minimal inhibitory concentration (MIC). The stock solutions of the compounds were diluted in DMSO to achieve concentrations ranging from 0.25 to 256 μg/mL. Plates were incubated at 35 °C and further read at 24 h (yeast) and 48 h (molds) by visual inspection. *Candida parapsilosis* (ATCC 22019) was used for quality control in all antifungal susceptibility tests. Amphotericin B was used as activity control and as a reference in order to compare the activity of acylhydroquinones. All assays were performed in triplicate. The cutoff values to determine susceptibility to antifungal drugs were based on CLSI rules. 

Both methodologies are detailed in CLSI guidelines M44-A2, M51-A, M27Ed4 and M38-A2 [45,46,47,48].

### 4.3. Antioxidant Capacity Assays 

#### 4.3.1. FRAP Assay

The FRAP assay was carried out as previously described [49] with some modifications. Briefly, the working solution was prepared by mixing 25 mL acetate buffer 0.3 mM (pH 3.4), 900 μL TPTZ solution, and 2.5 mL FeCl_3_ × 6H_2_O (20 mM) solution and then warmed at 37 °C before use. Compound **4** (10 μL) was allowed to react with 240 μL of the fresh FRAP solution and 20 μL of deionized water for 1 h in the dark. The absorbance was measured at 593 nm, using Tecan infinite M200 Pro (Männedorf, Switzerland). Finally, the standard curve was performed with the standard antioxidant Trolox^®^ at concentrations of 79, 119, 159, 199, 239, 279, 319, 359 and 399 μM. The inhibitory concentration (IC_50_) was calculated from the plot of inhibition percentage against the sample concentration and the result was expressed as mM of TEAC (Trolox^®^ equivalents)/mL of compound **4**. The determinations were performed in triplicate and are reported as mean values ± SD.

#### 4.3.2. ABTS^•+^ Free-Radical Scavenging Activity

The scavenging capacity to the ABTS radical of the compound **4** was calculated as described by Re et al. [50] with the following modifications: a solution of ABTS (7 mM) was prepared and mixed with a buffer solution of potassium persulfate. An aliquot (150 μL) of the solution was mixed with 12 μL of phosphate buffer and read at 734 nm using a Tecan infinite M200 Pro spectrophotometer. Then, a total of 10 µL of the compound **4** solution and 260 µL of the ABTS were incubated for 6 min and measured at the same wavelength. This experiment was triplicated using quercetin and as Trolox^®^ as controls. Afterwards, a curve of % ABTS^•+^ radical versus concentration was plotted and IC_50_ values were calculated. IC_50_ denotes the concentration of sample required to scavenge 50% of ABTS radical cation.

#### 4.3.3. DPPH• Free-Radical Scavenging Activity

Antioxidant scavenging activity was investigated according to a slightly modified DPPH assay [51], using quercetin and Trolox^®^ as controls. Compound **4** was dissolved with DMSO-MeOH mixture at a concentration of 10 mM. Then, from the stock solution, dilutions were prepared at concentrations of 0.25, 0.20, 0.15, 0.10 and 0.05 mM. To a total 250 μL of 2,2-diphenyl-1-picrylhydrazyl (DPPH) solution was added 20 µL of the compound **4**, and further incubated for 30 min at 37 °C. The experiment was carried out in triplicate and the absorbance was observed at a wavelength of 517 nm.

### 4.4. In Silico Simulations and ADMET Prediction

#### 4.4.1. Molecular Docking Calculations 

The Glide software from the Schrödinger suite [52] was used to study the binding modes of compound **4** into the active site of NADPH dehydrogenase using ligand–receptor molecular docking (flavonoids as CDK1 inhibitors: insights in their binding orientations and structure-activity relationship). To set up the docking experiments, water molecules in the active region were removed to allow each inhibitor to freely locate its optimum position inside the protein. Here, a 32 Å × 66 Å × 20 Å grid box covering the active site of NADPH dehydrogenase was used, centered on the co-crystalized inhibitor’s center of mass. The module LigPrep (LigPrep, Schrödinger 2017-1 LLC, BioLuminate, Schrödinger, New York, NY, USA) was used to assign ionization states and stereochemistry to the compounds. We used the Glide standard (SP) and extra-precision (XP) modes. To select the final pose, the binding mode with the lowest total docking energy (i.e., the more favorable pose) was chosen.

#### 4.4.2. Molecular Dynamic Simulations of Protein-Ligand Complex

In order to analyze the dynamic behavior between compound **4** and the NADPH dehydrogenase, we performed three replicas (of 100 ns MD simulations each) of this system. Here, the most stable protein–ligand complex obtained on the docking experiments was embedded in a water box with a buffer of 10 Å^3^ at neutral conditions. The SPC water model was used within the framework of the OPLS-AA force field [53]. MDS were performed using Desmond software [54]. Prior to the setup of routine MD simulations, the protein–ligand complex was minimized and pre-equilibrated using a relaxation routine implemented by default in Desmond. The program launched 9 steps composed of solute–solvent restrained minimizations and short MD relaxations (12 to 24 ps) that are used in order to stabilize the system. The van der Waals and electrostatic interaction cutoff was set to 9 Å. The temperature was kept at 300 °K by Nose-Hoover chain thermostat method.

#### 4.4.3. Free Energy Calculations

The Molecular Mechanics-Generalized Born Surface Area (MM-GBSA) approach, as implemented in Prime [29,55,56,57], was used to determine the binding-free energy (G_bind_) between both inhibitors and NADPH dehydrogenase (OYE1 wildtype). These extra computations were carried out in order to better understand the disparities in inhibitor-protein affinity. The calculations were performed on each inhibitor’s five best binding modes. The following equation was used to compute averaged G_bind_ values between the inhibitor and OYE1:$$\Delta G_{\mathrm{bind}}=\Delta E_{\mathrm{MM}}+\Delta G_{\mathrm{solv}}+\Delta G_{\mathrm{SA}}$$
where E_MM_ corresponds to the change in molecular mechanics (MM) energy in the gas phase due to binding, and comprises E_internal_ (bond, angle, and dihedral energies), E_elect_ (electrostatic), and E_vdw_ (van der Waals) energies. G_solv_ is the difference in GBSA solvation energy of the OYE1–inhibitor complex and the sum of the solvation energies for unliganded OYE1 and the inhibitor, while G_SA_ is the difference between the complex’s surface area energy and the sum of the unliganded OYE1 and inhibitor’s surface area energies.

#### 4.4.4. ADMET Prediction

The pkCSM online tool (http://biosig.unimelb.edu.au/pkcsm/prediction, accessed on 29 December 2021) [34] database was utilized to predict absorption, distribution, metabolism, excretion, and toxicity (ADMET) of compound **4** and amphotericin B as the control.

### 4.5. Statistical Analysis

GraphPad Prism 8.0.2 software (San Diego, CA, USA) was used for statistical analysis. The EC_50_ value (concentration of compound **4** causing half-maximal responses) was established by regression analysis.

## 5. Conclusions

A variety of 2-acylbenzo- and 2-acylnaphthohydroquinones, prepared through an environmentally friendly protocol, were evaluated for their antifungal activity against diverse *Candida* and filamentous fungi strains. From the antifungal screening, compound **4** appeared as the most active compound against *Candida* and filamentous fungi strains. These findings are of interest because some of the evaluated *Candida* strains are known to have decreased sensitivity to standard antifungal treatments. In the molecular docking study, compound **4** showed the best docking score on NADPH dehydrogenase and good stability during the molecular dynamics study carried out across 100 ns. In agreement with the ADMET prediction, the analysis made by using the pkCSM online tool revealed that compound **4** displays a good profile regarding absorption, distribution, metabolism, excretion, and toxicity properties. The results obtained in this work offer a rational basis for the potential therapeutic use of acylbenzohydroquinones compounds, leading to the development of new antifungal agents.

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
