# Peer review of "Antifungal Activity and In Silico Studies on 2-Acylated Benzo- and Naphthohydroquinones"

_molecules, 2022, doi:10.3390/molecules27093035_

Round 1

Reviewer 1 Report

The work on the “Antifungal Activity and In Silico Studies on 2-Acylated Benzo- 2 and Naphthohydroquinones” is a valuable contribution.

I would like to recommend the authors to re-write a lot of sentences due to the overlapping with the following article “Chemical Composition, In Vitro and In Silico Antioxidant Potential of Melissa officinalis subsp. officinalis Essential Oil”  The following lines should be edited :

18-20, 50-51, 112-119, 184-185, 232-233, 236-240, 310-311, and 339-343.

Abstract

  • The Authors did not mentioned any data regarding the antioxidant activities neither in title nor in abstract, why??
  • In the results I prefer to say better or more potent than don use “larger”
  • Line 30 it is better to use topical pharmacological use rather than topical biological use.
  • Introduction
  • Some typing errors were observed, extensive English revision are required
  • Control all the antifungal strins name they should be in italic
  • Line 52 we always use anticancer activities not anti-cancerous.
  • You can add the following recent publications for new isolated phenolic compounds from natural plants “https://doi.org/10.3390/molecules27031046” as antioxidants, and synthetic compounds containing phenols as antifungal “ DOI 10.1007/s13369-020-05114-0” and as antioxidant “https://doi.org/10.1038/s41598-022-07188-2”
  • Line 56-57 can you add the MIC values for mentioned compound ??
  • Can you edit “filamentous” in whole manuscript in the same type.
  • RESULTS
  • Result section need revision for data language and typing
  • Table 1 footnote has * and ndash but the main data in the table they are absent, I could not see the % of inhibition there, I think it is need editing
  • Line 84 “ glabrata was the most sensitive exhibiting the largest HOI against” largest what ??? diameters ?? or inhibition ?
  • Line 87 “more important than” I prefer more potent than
  • Line 89 “the only active against both” only active what ?? you have to add compounds
  • Line 104 you have to edit the verb phrase “the results show” Showed
  • Table 2 edit compound 12 results in the table it was shifted to right side and delete ndash from the footnote, as well as you mentioned the compounds structures in Table 1, so no need to repeat that again
  • Line 113 in vitro should be in Italic
  • The unit of IC50 values in Table 3 is it in µM or mM ??? please control them again
  • Did you test the positive controls with used binding pockets ??can you provide us with the docking score of compound 4 ??
  • Materials and Methods
  • Can you add the reagents which were used for the syntheses of these compounds regarding the reference?
  • Line 265-267 edit all strains names as in italic

Author Response

Reviewer 1

Comments and Suggestions for Authors

The work on the “Antifungal Activity and In Silico Studies on 2-Acylated Benzo- 2 and Naphthohydroquinones” is a valuable contribution.

 We thank the referee for his/her nice comment.

I would like to recommend the authors to re-write a lot of sentences due to the overlapping with the following article “Chemical Composition, In Vitro and In Silico Antioxidant Potential of Melissa officinalis subsp. officinalis Essential Oil”  The following lines should be edited :

18-20, 50-51, 112-119, 184-185, 232-233, 236-240, 310-311, and 339-343.

The indicated lines were rewritten as suggested by the referee.

 Abstract

  • The Authors did not mentioned any data regarding the antioxidant activities neither in title nor in abstract, why??.

Because antioxidant activity was only tested with regard to compound 4. Therefore it’s not justified to include it in the tittle (several hydroquinones were studied). Anyway it was included in the abstract.

  • In the results I prefer to say better or more potent than don use “larger”
  • We agree. It was modified.
  • Line 30 it is better to use topical pharmacological use rather than topical biological use. We agree. It was modified.

Introduction

  • Some typing errors were observed, extensive English revision are required.

Mistakes and errors were corrected.

  • Control all the antifungal strins name they should be in italic.

It was done.

  • Line 52 we always use anticancer activities not anti-cancerous.

It was corrected.

  • You can add the following recent publications for new isolated phenolic compounds from natural plants “https://doi.org/10.3390/molecules27031046” as antioxidants, and synthetic compounds containing phenols as antifungal “ DOI 10.1007/s13369-020-05114-0” and as antioxidant https://doi.org/10.1038/s41598-022-07188-2 .

We thank the referee for such suggestion. These references are now included in the revised version.

  • Line 56-57 can you add the MIC values for mentioned compound??

No, because in the quoted reference, the assay was not made to calculated MIC, but using other way to express antifungal activity.

  • Can you edit “filamentous” in whole manuscript in the same type.

It was done.

RESULTS

  • Result section need revision for data language and typing.

It was revised.

  • Table 1 footnote has * and ndash but the main data in the table they are absent, I could not see the % of inhibition there, I think it is need editing.

It was corrected.

  • Line 84 “ glabrata was the most sensitive exhibiting the largest HOI against” largest what ??? diameters ?? or inhibition ?

 It’s now corrected and best explained in the revised text.

  • Line 87 “more important than” I prefer more potent than.

We agree. It was corrected.

  • Line 89 “the only active against both” only active what ?? you have to add compounds. This sentence was rewritten.
  • Line 104 you have to edit the verb phrase “the results show” Showed

It was corrected.

  • Table 2 edit compound 12 results in the table it was shifted to right side and delete ndash from the footnote, as well as you mentioned the compounds structures in Table 1, so no need to repeat that again

 It was corrected.

  • Line 113 in vitro should be in Italic

It was done

  • The unit of IC50 values in Table 3 is it in µM or mM ??? please control them again.

Sorry for this confusion. It is mM.

  • Did you test the positive controls with used binding pockets ??can you provide us with the docking score of compound 4 ??

Its glide score docking was -4.481 Kcal/mol. Regarding amphotericin B the assay calculation was not conducted, because its mechanism of action has been clearly demonstrated to be mediated by an interaction with Ergosterol.

  • Materials and Methods
  • Can you add the reagents which were used for the syntheses of these compounds regarding the reference?

It was done.

  • Line 265-267 edit all strains names as in italic

 It was done.

Reviewer 2 Report

Please see the following comments and revise the manuscript accordingly. 

- Please be consistent in writing organism names. The names should be written in full throughout the manuscript or written in full for the first time and afterwards, in abbreviated form. In the manuscript these rules have not been followed.

Sample errors: L41 (section introduction; C. albicans), L265 (section methods; Candida albicans)

L55: C. cladosporioides and C. sphaerospermum have not been previously written in full form.

Several similar errors.

-Same problem with other words e.g. HOI

- All organism names should be italicized. E.g. L265-267

- L269: “Sabouraud agar (PDA)”: What do you mean by (PDA)? It is a common abbreviation in Mycology stands for Potato Dextrose Agar which is different from Sabouraud agar. If it is the name for manufacturer, please write it in full form to avoid confusion, otherwise consider editing.

-L77: Candida spp are not mold. Please correct.

-In section results two words “species” and “strain” have been used incorrectly. There are four filamentous species (A. fumigatus, A. terreus. A. kiliense, R. oryzae) while two strains from A. fumigatus species are included. So, “while only three mol-82 ecules were active against 4-5 species of Filamentous fungi.” In L82-83 is incorrect because you have not included 5 species.

Author Response

Reviewer 2

Comments and Suggestions for Authors

Please see the following comments and revise the manuscript accordingly. 

We did all changes according to suggestions made by the referee.

- Please be consistent in writing organism names. The names should be written in full throughout the manuscript or written in full for the first time and afterwards, in abbreviated form. In the manuscript these rules have not been followed.

It was corrected.

Sample errors: L41 (section introduction; C. albicans), L265 (section methods; Candida albicans).

It was done.

L55: C. cladosporioides and C. sphaerospermum have not been previously written in full form.

It was done.

Several similar errors.

-Same problem with other words e.g. HOI.

It was corrected

- All organism names should be italicized. E.g. L265-267.

It was done.

- L269: “Sabouraud agar (PDA)”: What do you mean by (PDA)? It is a common abbreviation in Mycology stands for Potato Dextrose Agar which is different from Sabouraud agar. If it is the name for manufacturer, please write it in full form to avoid confusion, otherwise consider editing.

We regret such mistake. It was SDA (Sabouraud Dextrose Agar)

-L77: Candida spp are not mold. Please correct.

It was corrected.

-In section results two words “species” and “strain” have been used incorrectly. There are four filamentous species (A. fumigatus, A. terreus. A. kiliense, R. oryzae) while two strains from A. fumigatus species are included. So, “while only three molecules were active against 4-5 species of Filamentous fungi.” In L82-83 is incorrect because you have not included 5 species.

 This sentence was rewritten.

Reviewer 3 Report

see attachments

Author Response

Reviewer 3

The paper by Ríos et al. deals with the topical issue of the search for new antifungals to overcome the problems of toxicity and spread of antimicrobial resistance. A careful revision of English language should be performed throughout the text. Some suggestions for corrections are provided directly on the pdf version of the manuscript in attachment.

We thank the comments of the referee. His/her criticisms were taken into account in this revised version. 

General issues

The authors could better explain how this manuscript leads to an advancement of knowledge, compared to other papers (cited) on the topic of phenolic compounds endowed with antifungal activity, and the rationale (if any) for the selection of the synthesized acylhydroquinone derivatives. We think our results contribute to knowledgement in this field. For example, a potential intracellular target (OYE1 enzyme) was identified to explain antifungal activity. The rationale to select the acylhydroquinones was made due to their structural relationship with known molecules displaying antifungal activity. Indeed they are phenolic compounds in which the activity may be modulated by coupling several substituent groups.    

Line 52: authors report an “antioxidant [14-16] activity” for phenolic compounds, but the cited papers rather refer to redox-active compounds that target cellular antioxidation systems.

According to the referee, the references [14-16] were changed. We included now the following articles.

Grati, W.; Samet, S.; Bouzayani, B.; Ayachi, A.; Treilhou, M.; Téné, N.; Mezghani-Jarraya, R. HESI-MS/MS Analysis of Phenolic Compounds from Calendula aegyptiaca Fruits Extracts and Evaluation of Their Antioxidant Activities. Molecules. 2022, 27, 2314-2327.

Adjdir, S.; Benariba, N.; El Haci, I. A.; Ouffai, K.;  Bekkara, F. A.; Djaziri, R. Antioxidant activity and phenolic compounds identification of Micromeria inodora (Desf.) Benth. from Western Algeria. Nat. Prod. Res. 2021, 35, 2963-2966. 

Wang, M.; Jiang, N.; Wang, Y.; Jiang, D.; Feng, X. Characterization of Phenolic Compounds from Early and Late Ripening Sweet Cherries and Their Antioxidant and Antifungal Activities. J. Agric. Food. Chem. 2017, 65, 5413-5420. 

While, in light of the above references, it is understandable the choice to examine, by molecular docking, the binding modes of compound 4 to the OYE1 enzyme, it is not clear why they have chosen to evaluate the antioxidant capacity of compound 4 itself. This issue needs a clarification.

As explained in point 2.4 the interaction of compound 4 with OYE1 NADPH dehydrogenase involved redox reactions. Subsequently, and due to its phenolic nature, it is appeared rather logic to measure antioxidant activity in compound 4

Beyond ADMET prediction, in vitro evaluation of toxicity of the most promising compound (e.g., hemolysis, cytotoxicity against cultured mammalian cells) should be performed as a preliminary screening.

We agreed with the referee but the basic idea in this paper was to identify antifungal activity. Additional assays are required to further develop drugs.

Specific issues

Filamentous fungi throughout the text should NOT be in italics: filamentous fungi (or molds).

Microorganisms (genus and species) should always be in italics: check throughout the text, including References section.

Many references in Introduction section are not up to date.

Table 1 Footnote: why % growth inhibition? Isn’t the Halo of Inhibition measured in mm? Moreover, with reference to this test, it is not clear how the determination has been carried out. In line 77 it is stated: “…after 48 h exposition to acylhydroquinones at 50 ppm.”, while in Materials and Methods section 4.2.2. Antifungal susceptibility testing, it is stated: “…Given that disk diffusion tests are qualitative, concentrations of the stock solutions were prepared at the rate of 1280 μg/mL.” The authors should clarify how the disk diffusion test was performed and interpreted, and the concentration of the compounds tested. It should be also specified if a control constituted by diluent (DMSO) alone was used.

It is highlighted in both Abstract (line 31) and Conclusions (line 371): “…some of the evaluated Candida strains are multi-resistant…” but it is not specified which strains and which antifungals the authors refer to. Please clarify (e.g., in Materials and Methods section).

All corrections regarding these specific issues were corrected and included in the revised version.

Round 2

Reviewer 1 Report

almost all comments were answered accordingly, well done work 

Reviewer 3 Report

see attachment
